# A point-of-care diagnostic for drug-induced liver injury using surface-enhanced Raman scattering lateral flow immunoassay

Sian Sloan-Dennison[1,7], Kathleen M. Scullion[2,7], Benjamin Clark[1,7], Paul Fineran[3], Joanne Mair[3], Stacey Laing[1], Neil C. Shand[4], Cicely Rathmell[5], David Creasey[5], Dieter Bingemann [5], Jonathan Faircloth[5], Mark Zieg [5], Elizabeth Varghese[6], Christopher J. Weir [6], James W. Dear [2,8] ✉, Karen Faulds[1,8] & Duncan Graham [1,8]

Paracetamol overdose (POD) is common, with approximately 100,000 cases attending UK hospitals annually. Timely antidote administration is crucial for patients at risk of developing acute liver failure. A rapid point-of-care (POC) assay is required to identify high-risk patients with fit-for-purpose sensitivity and specificity. Here we show that by measuring a circulating biomarker, cytokeratin-18 (K18), accurate detection of drug-induced liver injury (DILI) is possible. To achieve this, we created an in vitro diagnostic medical device designed to quantitatively detect serum K18, consisting of a Lateral Flow Immunoassay (LFIA) paired with a bespoke handheld Raman Reader (HRR) to produce quantitative surface-enhanced Raman scattering (SERS). The diagnostic was assessed in 2 performance evaluation studies using 199 serum samples from individuals following POD. The device achieves diagnostic accuracy for DILI with a specificity of 94% and sensitivity of 82%. Here we show that SERS-LFIA can rapidly identify patients with DILI, allowing individualised treatment pathways.

Paracetamol (acetaminophen) is one of the most widely used analgesics globally. When taken at the recommended therapeutic dose, it is believed to have an excellent safety profile[1,2]. However, in overdose, paracetamol is the commonest cause of acute liver failure (ALF) in the USA and Europe. It is estimated that there are at least 1000 paracetamol overdose (POD)-ALF cases treated in Liver Transplantation Units in the US and EU (mostly UK) each year. This group has a mortality rate of ~30–35%, with around 30% receiving a liver transplant. Most transplant recipients survive, but will require continuous lifelong treatment and frequently experience complications related to the immunosuppressive medications necessary for transplantation[3]. POD is common, with around 100,000 people presenting to emergency departments following a POD and ~50,000 patients requiring emergency antidote treatment to prevent drug-induced liver injury (DILI), and subsequent ALF, every year in the UK alone[4]. In 2021, US poison centres received more than 80,000 cases involving a paracetamol product[5]. Treatment with the antidote, N-acetylcysteine (NAC), is highly effective when administered within 8 h of overdose[6]. However, NAC offers minimal benefits if treatment is delayed for more than around 20 h.

[1]Department of Pure and Applied Chemistry, Technology and Innovation Centre, University of Strathclyde, Glasgow, UK. [2]Centre for Cardiovascular Science, The Queen's Medical Research Institute, University of Edinburgh, Edinburgh, UK. [3]Translational Healthcare Technologies Group, Centre for Inflammation Research, Institute for Regeneration and Repair, Edinburgh, UK. [4]Defence Science and Technology Laboratory (DSTL), Porton Down, Salisbury, UK. [5]Wasatch Photonics, Morrisville, NC, USA. [6]Edinburgh Clinical Trials Unit, Usher Institute, University of Edinburgh, Edinburgh, UK. [7]These authors contributed equally: Sian Sloan-Dennison, Kathleen M. Scullion, Benjamin Clark. [8]These authors jointly supervised this work: James W. Dear, Karen Faulds, Duncan Graham. ✉e-mail: james.dear@ed.ac.uk

There are no specific symptoms or clinical signs of early liver damage and, therefore, clinicians rely on blood 'liver functions tests' to pick up liver injury after POD. These tests are performed in central hospital laboratories, which results in a time delay between blood sampling and therapeutic decision making. Furthermore, the standard serum biomarker for DILI diagnosis, alanine aminotransferase (ALT) activity, increases too slowly post-POD to accurately diagnose DILI within the NAC optimal therapeutic window. Therefore, a rapid point-of-care (POC) assay is needed to identify high-risk patients with fit-for-purpose sensitivity and specificity to enhance early targeted NAC treatment.

Cytokeratin 18 (K18) is a mechanistic biomarker of liver injury that provides information about the type of cell death. The caspase-cleaved form of K18 (cc-K18) is released early during cellular structural rearrangement and apoptosis. The full-length form of K18 is passively released upon cell necrosis[7]. Multiple studies have demonstrated that K18 is a sensitive and specific biomarker that can accurately distinguish patients with and without DILI at an earlier time point than the gold standard, ALT[8,9]. K18 has regulatory support for use as a biomarker in clinical trials from the US Food and Drug Administration. K18 has potential utility for predicting DILI and prognostic assessment of outcome, further emphasising its potential as a promising DILI biomarker[10,11].

The gold standard method for K18 quantification is an enzyme-linked immunosorbent assay (ELISA). Although the ELISA produces accurate and quantitative results, the process takes several hours to perform by trained staff using specialist equipment and costly materials[12]. In clinical practice, the results would take too long, delaying NAC treatment beyond the optimal therapeutic window of 8 h. To maximise the benefit that K18 offers, a rapid and quantitative test must be developed. The test should be capable of being used in the hospital Emergency Department at the POC, with minimal training required, and be relatively cheap to produce. A suitable tool to meet this product profile is a lateral flow immunoassay (LFIA). LFIAs are paper-based tests performed in a plastic cassette that are designed to detect a biomarker of interest. Capture antibodies labelled with gold nanoparticles form sandwich immunoassays with detection antibodies on the test line when the biomarker is present in a sample, immobilising them and producing a red line. Conventionally, they provide a binary visual result based on the presence or absence of a test line, akin to the SARS-CoV-2 LFIA test. However, when quantification of the biomarker concentration is required for diagnosis, monitoring treatment, improving patient stratification, or if the concentration is very low, the LFIA can be combined with surface-enhanced Raman scattering (SERS).

SERS is a vibrational spectroscopy technique that enhances the Raman scattering of molecules that are bound to a roughened metal surface via the excitation of the surface plasmons. Typically, gold and silver nanoparticles have been used to provide enhancement factors of up to $10^{10}$[13]. To apply SERS as a read-out technique for an LFIA, nanoparticles are labelled with a Raman reporter molecule, as well as an antibody[14]. When the LFIA is run, and the sandwich immunoassay has formed on the test line, the line can be analysed using a Raman spectrometer. The resulting SERS spectrum will be that of the Raman reporter bound to the immobilised nanoparticle. As SERS is quantitative and increases in relation to the number of nanoparticles and Raman reporters present, the intensity of the SERS spectrum can be related to the biomarker concentration[15].

The coupling of SERS and LFIAs has been used to detect low concentrations of different biomarkers. Most of the previous research has focused on developing nanoparticles that produce strong SERS signals. By modifying the shape, metals and Raman reporters, limits of detection in the pg/mL range have been achieved for a variety of clinical biomarkers, including pneumolysin, interleukin-6 (IL6) and HIV-1 DNA[16]. However, despite the high levels of sensitivity, the SERS of the test lines were analysed on large benchtop Raman readers attached to microscopes, and Raman mapping the test lines can take up to 20 min per sample[17]. This methodology is not feasible in a POC setting, and, therefore, the development of small and portable Raman readers is required to allow SERS-LFIA to be utilised in a clinical environment[18].

In this work, we create, evaluate and refine a SERS-LFIA coupled to a bespoke handheld Raman reader (HRR), specifically designed for LFIA measurements. This allows us to quantify a biomarker, K18, to identify POD patients at an increased risk of developing DILI.

## Results

### Creation of a bespoke diagnostic assay and reader

To detect and quantify both full-length and cc-K18 in patient samples, a SERS-LFIA strip was created and coupled with a bespoke HRR to produce a quantitative output. The accuracy, sensitivity and specificity of the test for DILI detection were assessed in two diagnostic performance tests (study A and B). The SERS-LFIA strip contained conjugates, consisting of gold nanoparticles (AuNP) coated in a Raman reporter and antibodies (Ab). The AuNPs were synthesised via a citrate reduction method, functionalised with the Raman reporter 4,4'-dipyridyl (DIPY) and encapsulated in a silica shell ($SiO_2$). The silica shell was then functionalised with antibodies specific to K18, as well as bovine serum albumin (BSA), which increases protection and reduces non-specific binding. The resulting conjugate is called 'Au-DIPY-$SiO_2$-Ab NP'. Characterisation data are presented in Supplementary Figs. 1 and 2 and Supplementary Table 1. The LFIA strip consisted of treated sample pads, the Au-DIPY-$SiO_2$-Ab NP conjugate pad, test and control lines and a collection pad. The strip was then encased in a 3D-printed cassette (Fig. 1A). To run a sample on the SERS-LFIA, serum was diluted in running buffer and dispensed onto the sample port of the cassette. It was left to run for 30 min then analysed on a bespoke HRR.

To use the HRR in a hospital Emergency Department, it should be operated as a Class 3R for safety, meaning that the laser must have a maximum output power of 5 mW[19,20]. This was considered, and the HRR was designed to capture as many scattered photons from the test and control line as possible to optimise the sensitivity at a low laser power. To achieve this, a large numerical aperture (f/1.1) was used, which allowed more photons to be collected. The miniaturised optical bench of the HRR was based around a custom volume phase holographic transmission grating optimised for high efficiency and minimal scatter (Wasatch Photonics LTD[21]), matched with an uncooled, commercial line-array sensor to achieve ultra-high sensitivity in a handheld diagnostic instrument that could be cost-effectively manufactured in volume. To optimise the interface with the LFIA, the laser was projected as a line onto the test line using a Powell lens. This increased the area of the test line that could be interrogated and reduced the number of measurements per sample. Therefore, the value of this HRR lies in three elements: (1) use of a highly efficient f/1.1 transmissive design with a low-cost Complementary Metal-Oxide Semiconductor detector for enhanced sensitivity, (2) inclusion of a Powell lens to maximise overlap between the excitation laser and LFIA strip lines, with consequent direct imaging onto the detector, and (3) use of a Class 3R laser to effectively and repeatably perform SERS measurements on a lateral flow strip, thus reducing laser exposure risk to facilitate implementation in clinical settings. A detailed optical schema of the HRR is shown in Supplementary Fig. 3.

Studies A and B used different iterations of the HRR, consisting of two slightly different coupling methods to the SERS-LFIA strip (Fig. 1B). In study A, the HRR was coupled to the SERS-LFIA using a 3D-printed accessory that attached via magnets in front of the laser aperture (HRR 3A). The accessory was designed to hold the SERS-LFIA at the correct focal distance and allowed the user to move the SERS-LFIA strip back and forth across the laser line, to achieve the optimum overlap, and thus maximise the SERS signal from the strip. In study B, the coupling of the SERS-LFIA to the HRR was modified and enclosed to increase the usability and safety of the HRR (HRR 4A). Cassette sleeves were

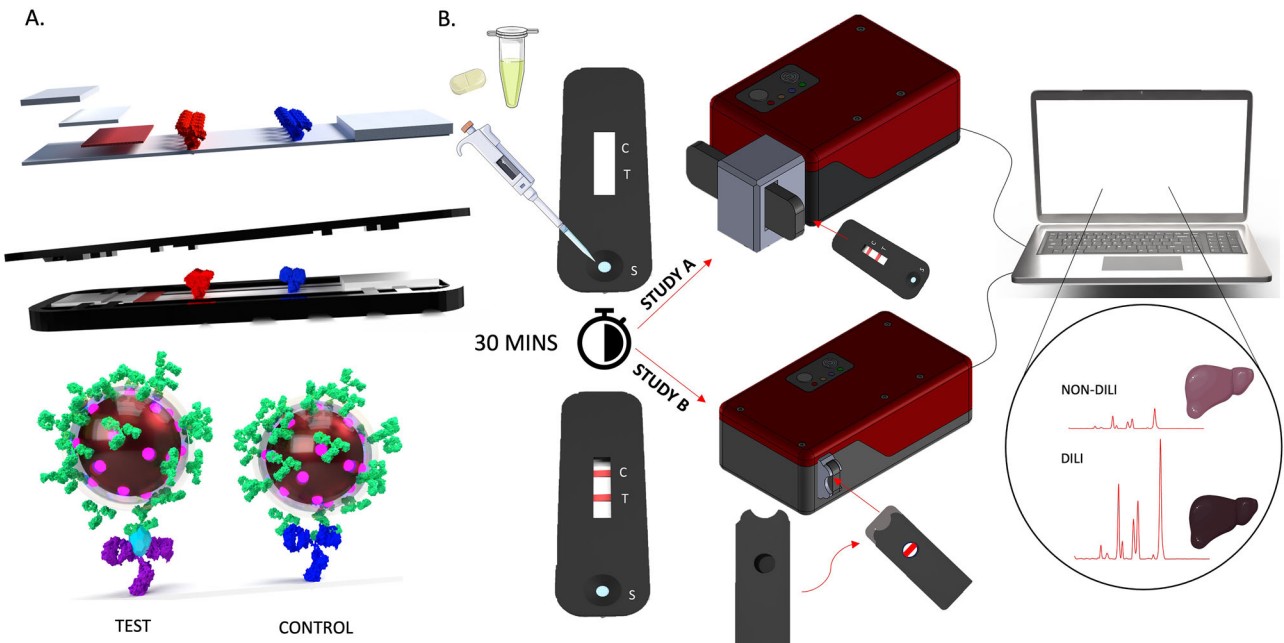

**Fig. 1 | Pre-clinical SERS-LFIA development. A** A schematic illustrating SERS-LFIA strip architecture showing the location of the test (red) and control (blue) antibody lines, the strip housed in a 3D-printed cassette, and the immunoassays that form on the test and control line when K18 is present. **B** The SERS-LFIA procedure. Serum collected from a patient was diluted in running buffer and pipetted onto the sample port of the cassette. The test is allowed to run for 20 min and analysed with the HRR. In study A, the cassette was inserted into the 3D-printed accessory and manually lined up with the laser (HRR 3A, top). In study B, the cassette was slotted into the cassette sleeve and inserted into a built-in slot within the HRR (HRR 4A, bottom). The sleeve aided alignment of the line with the laser. The HRR units were connected to a laptop, and the intensity of the SERS signal of the test and control line determined whether the patient had developed DILI. Technical illustrations were generated using SolidWorks CAD software and Adobe Illustrator. Figure contains a modified illustration from Servier Medical Art, licensed under a Creative Commons Attribution 4.0 International License (CC BY 4.0). HRR handheld Raman reader, K18 cytokeratin 18, LFIA lateral flow immunoassay, SERS surface-enhanced Raman scattering.

designed to ensure that when the SERS-LFIA cassette was slotted inside, only the test line was visible through the circular window. The sleave was then inserted into a slot built into the instrument, allowing the test line to be at the correct focal distance and position in front of the laser (Fig. 1B). In practice, the interface between the laser and sample are fully enclosed in HRR 3A and 4A, effectively rendering the devices as Class 1, as it meets the definition 'Class 1 lasers have low radiated power or are enclosed to prevent radiation from escaping'. However, as we are without certification, we have classed them both as 3R. Each HRR was connected to a laptop, and the intensity of the test line relative to the control line, based on the intensity of the DIPY Raman reporter peak at 1612 cm⁻¹ was used to infer DILI status.

To determine the concentration of K18 in an unknown patient sample, a calibration curve was produced to include clinically relevant concentrations of K18 in serum, based on reference ranges described by Church et al.[22]. The K18 concentration upper limit of normal (ULN) for healthy volunteers was between 7.3 and 9.1 ng/mL. The geometric mean for DILI was 81.5 ng/mL for patients who survived/did not require a transplant 6 months post-DILI onset and 628 ng/mL for those who died/required a transplant. We used a range from 0 to 750 ng/mL to reflect these values. Representative images of the SERS-LFIA tests and resulting SERS measurements used as part of Study A are presented in Supplementary Figs. 4–6, and the results of the calibration curve developed as part of Study B are presented in Fig. 2. Healthy serum spiked at a range of K18 concentrations was applied to the SERS-LFIA and measured with the HRR. The visual output is presented in Fig. 2A, where only the control line is present for serum that is not spiked with K18. A gradual increase in the visual intensity of the lower test line is observed with increasing K18 concentrations. When the SERS-LFIA was coupled with the HRR, and the SERS signals from the lines were measured, there was greater SERS intensity measured at higher K18

concentrations (Fig. 2B). To reduce variation, the control line was also measured, and pixel-by-pixel linear regression of the test spectrum versus the control spectrum was used to standardise the SERS output. Additional information detailing the calibration curve created using the SERS intensity obtained from the test line alone versus linear regression is detailed in Supplementary Figs. 4–6. The linear regression analysis produced a SERS slope output which demonstrated a linear relationship with the spiked K18 concentrations in the samples ($R^2 = 0.98$, Fig. 2C). There was also a correlation between the SERS slope measured compared to the gold standard ELISA measurement for K18, demonstrating a similar linear gradient for both calibration graphs (Fig. 2D).

### Evaluation of the diagnostic assay

We carried out two diagnostic performance evaluation studies using serum samples from the Markers and Paracetamol Poisoning Study 2 (MAPP2) trial. The multiple operators of the SERS-LFIA test were blinded to the status of the samples (non-DILI or DILI). Results were analysed by an independent statistician as per a pre-defined statistical analysis plan. Full details of the clinical studies are reported as per the Standards for Reporting of Diagnostic Accuracy Studies guidelines in the supplemental information (Supplementary Table 4).

The total combined number of patients included in the retrospective case-control studies was 199 ($n = 99$ for the initial study A and $n = 100$ for study B). The demographics and clinical characteristics for the patients included in the studies are presented in Table 1. There were no substantial differences between the patients included in study A and study B. In study A, each sample was analysed in triplicate by three independent operators to assess the reproducibility of the SERS-LFIA and usability of the HRR. In study B, each sample was analysed once by a single operator as modifications to the HRR addressed

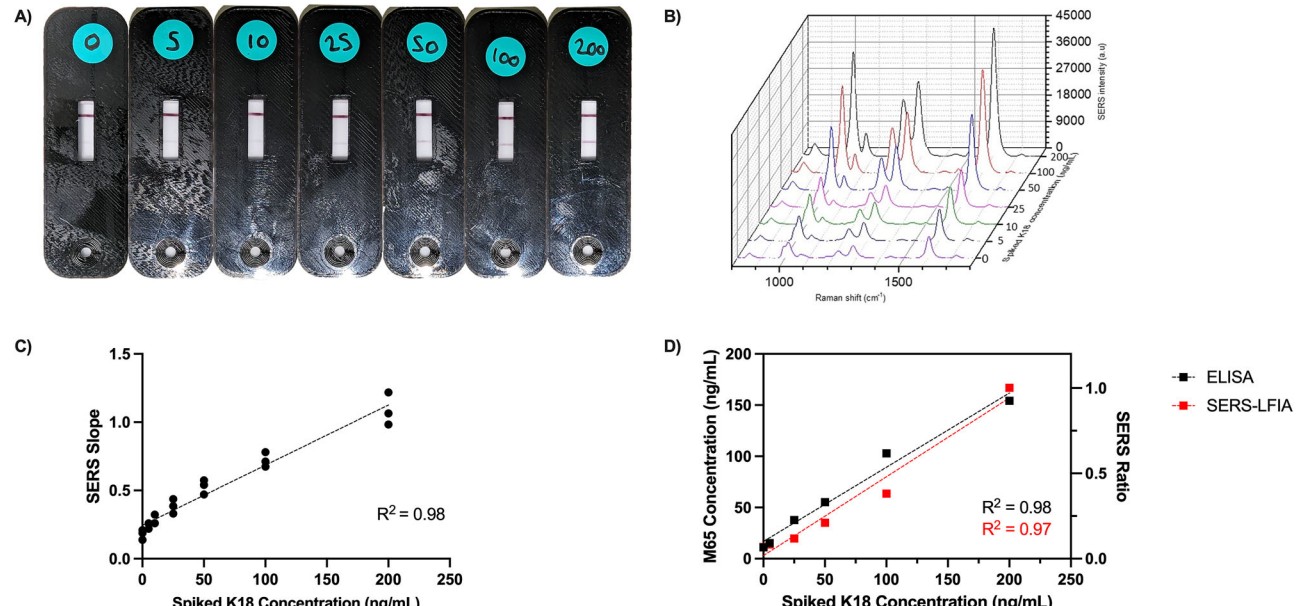

**Fig. 2 | Development of LFIA in combination with a handheld Raman reader (HRR) for use with clinical samples. A** Representative images of the LFIA calibration curve produced using spiked serum samples at clinically relevant concentrations. **B** Representative SERS spectra for the calibration curve. SERS spectra collected using HRR 4A with 785 nm laser excitation, 3.5 mW laser power, 1 s acquisition and 5 scanning averages. **C** SERS slope for calibration curve using 3 independent donors. **D** K18 concentration measured via M65 ELISA and LFIA with SERS presented as mean values. ELISA enzyme-linked immunosorbent assay, K18 cytokeratin 18, LFIA lateral flow immunoassay, SERS surface-enhanced Raman scattering.

variation in measurements. The K18 concentration was measured with the SERS-LFIA test in serum samples from patients presenting to the Emergency Department following a POD. This included patients who did not develop DILI (non-DILI) and those who did develop DILI. DILI was defined as an ALT elevation of ≥5 times the ULN[23].

For study A, the pre-defined primary statistical output was the K18 concentration, determined by a randomly selected first SERS measurement, from each of the 3 independent operators (Fig. 3A, B, see 'Methods' for details). Using the linear regression of the SERS measurements from the HRR and the calibration curve (Supplementary Fig. 6), the median concentration (IQR) of K18 was determined to be higher in the DILI cohort than the non-DILI cohort (DILI 161.0 ng/mL, 103.5–226.5 ng/mL V, non-DILI 41.0 ng/mL, 27.8–62.8 ng/mL, ROC-AUC = 0.93, 95% CI 0.88–0.98). The secondary statistical output was the geometric mean of all the K18 measurements obtained from each of the operators (Fig. 3C, D). The geometric mean (95% CI) K18 concentration for the analysers was higher in the DILI cohort than the non-DILI cohort (DILI 146.2 ng/mL (125.0–171.0 ng/mL), V non-DILI 41.3 ng/mL, 35.3–48.3 ng/mL, ROC-AUC = 0.95, 95% CI 0.91–0.99). Comparison of the concentrations of ALT, ELISA K18 and SERS-LFIA K18 from a selection of samples is presented in Supplementary Fig. 7. The data for the individual analysers are detailed in Supplementary Table 2. Disaggregated gender data are presented for studies A and B (Supplementary Fig. 8). Based on the data obtained in study A and feedback from the operators, the coupling between the SERS-LFIA strip and HRR was adapted to improve the user experience. The SERS-LFIA test, with integrated coupling between the LFIA strip and the HRR (HRR 4A) was then evaluated with serum samples in a second cohort of POD patients, study B (Fig. 3E, F), using the slope of the SERS measurements from the HRR and the calibration curve (Fig. 2C). With the refined device, used in study B, the median concentration of K18 was higher in the DILI cohort than the non-DILI cohort (DILI 213.2 ng/mL, 149.2–261.7 ng/mL V non-DILI 55.6 ng/mL, 37.2–81.4 ng/mL, ROC-AUC = 0.97, 95% CI 0.94–1.0).

The participant flow and assay accuracy for studies A and B (for SERS analysis) are presented in Fig. 4 and Table 2. The K18

concentrations selected for the primary analysis of Study A, using a randomly selected first measurement from one of the analysers, and secondary analysis, using the geometric mean of the K18 measurement by each of the three analysers, were defined as: (1) the cut-point which had the highest combination of sensitivity and specificity from the cut-points with close to 95% specificity (primary–85 ng/mL, secondary–91 ng/mL) and (2) the cut-point which had the highest combination of specificity and sensitivity from the cut-points with close to 95% sensitivity (primary–49 ng/mL, secondary–58 ng/mL). Study B cut-points were determined for the mean values obtained by the analyser with the same classifications as Study A, cut-points determined as (1) 132 ng/mL and (2) 108 ng/mL.

For comparison with the SERS data, visual analysis of the LFIA was performed using a pre-defined scoring system developed with reference images (Supplementary Fig. 9). Three independent reviewers were blinded to the DILI status and scored the LFIAs based on test and control line intensity. The LFIA images were classified as positive, weakly positive, or negative. Using the visual scoring system, diagnostic accuracy improved for multiple measurements following refinement of the devices (study A: 71.0% sensitivity, 77.0% specificity, 73.0% accuracy vs study B: 96.7% sensitivity, 58.8% specificity, 77.9% accuracy). Therefore, the higher sensitivity also allowed for improved identification of DILI patients by binary visual assessment; however, specificity was reduced as a result.

Agreement for the quantification of K18 in the clinical samples was compared between SERS-LFIA and K18 (M65) ELISA. Bland-Altman plots are presented for study A and B in Fig. 5. In both studies, the ratio between the two methods is lower for DILI than non-DILI (study A DILI = 0.5, non-DILI = 3.8 and study B DILI = 1.0 and non-DILI = 4.7). The non-DILI group also had a greater ratio between the upper and lower limits of agreement in comparison to DILI for both study A and B. The differences between the log-transformed K18 measurements using SERS and ELISA have a reasonably normal distribution, but as there is evidence of an association between the difference in the log-transformed K18 measurements and the mean of the log-transformed K18 measurements using SERS and ELISA, the estimates

**Table 1 | Demographics and clinical results for patients included in studies A and B**

| | Study A | | Study B | |
|---|---|---|---|---|
| | **Non-DILI (n = 50)** | **DILI (n = 49)** | **Non-DILI (n = 50)** | **DILI (n = 50)** |
| **Age years: Median (IQR)** | 20.0 (18.0–30.0) | 42.0 (19.5–48.0) | 30.0 (20.0–43.3) | 31.5 (21.8–46.5) |
| **Gender: Male (%)** | 9 (18) | 20 (41) | 12 (24) | 26 (52) |
| **Ethnicity (%)** | | | | |
| White (British) | 7 (14) | 0 | 15 (30) | 2 (4) |
| White (English) | 3 (6) | 0 | 2 (4) | 0 |
| White (Scottish) | 38 (76) | 44 (90) | 31 (62) | 42 (84) |
| White (Irish) | 0 | 0 | 0 | 2 (4) |
| White (Other) | 1 (2) | 3 (6) | 1 (2) | 2 (4) |
| Mixed (White and Black) | 0 | 1 (2) | 0 | 1 (2) |
| Mixed (Other) | 0 | 0 | 1 (2) | 0 |
| Asian or British Asian (Indian) | 0 | 1 (2) | 0 | 0 |
| Other | 1 (2) | 0 | 0 | 1 (2) |
| **Overdose type (%)** | | | | |
| Acute overdose > 8 h | 5 (10) | 11 (22) | 6 (12) | 21 (42) |
| Acute overdose < 8 h | 24 (48) | 17 (35) | 33 (66) | 10 (20) |
| Supra-therapeutic overdose | 5 (10) | 14 (29) | 2 (4) | 11 (22) |
| Staggered intentional overdose | 16 (32) | 6 (12) | 8 (16) | 8 (16) |
| Unknown | 0 | 1 (2) | 1 (2) | 0 |
| **Total paracetamol ingested in grams: Median (IQR)** | 14.0 (8.0–20.0) | 30.0 (17.0–45.0) | 16.0 (8.8–23.5) | 29.5 (16.0–48.1) |
| **Ingestion of other drugs: Yes (%)** | 26 (52) | 22 (45) | 34 (68) | 30 (60) |
| **Anti-coagulants** | 1 (2) | 1 (2) | 0 | 0 |
| **Non-opioid analgesics** | 2 (4) | 0 | 2 (4) | 3 (6) |
| **NSAIDs** | 11 (22) | 5 (10) | 13 (26) | 5 (10) |
| **Cardiovascular drugs** | 1 (2) | 0 | 0 | 0 |
| **Alcohol** | 9 (18) | 7 (14) | 9 (18) | 15 (30) |
| **Opioids** | 4 (8) | 10 (20) | 9 (18) | 10 (20) |
| **SSRIs** | 1 (2) | 2 (4) | 4 (8) | 2 (4) |
| **Tricyclic antidepressants** | 1 (2) | 0 | 4 (8) | 1 (2) |
| **Benzodiazepines** | 5 (10) | 1 (2) | 4 (8) | 0 |
| **Other** | 8 (16) | 8 (16) | 17 (34) | 6 (12) |
| **Prothrombin (secs): Median (IQR)** | 14.0 (13.0–16.0) | 18.1 (14.1–23.8) | 13.0 (11.0–14.0) | 18.0 (14.1–22.7) |
| **ALP (U/L): Median (IQR)** | 65.5 (55.0–71.0) | 89.0 (68.5–107.0) | 74.0 (60.0–88.5) | 77.0 (58.0–111.5) |
| **Creatinine (µmol/L): Median (IQR)** | 60.5 (53.0–66.3) | 68.0 (55.8–78.3) | 64.0 (59.0 74.0) | 62.5 (53.3–76.8) |
| **Urea (mmol/L): Median (IQR)** | 2.7 (2.3–3.3) | 4.1 (3.0–6.1) | 3.5 (2.7–4.6) | 3.7 (2.4–5.1) |
| **WBC (x10⁹/L): Median (IQR)** | 7.8 (6.0–8.9) | 7.9 (5.2–10.3) | 8.8 (6.7–10.8) | 8.1 (6.4–9.6) |
| **Potassium (mmol/L): Median (IQR)** | 3.9 (3.7–4.0) | 3.7 (3.4–4.1) | 3.9 (3.6–4.0) | 3.6 (3.4–3.8) |
| **INR: Median (IQR)** | 1.2 (1.1–1.3) | 1.7 (1.4–2.1) | 1.1 (1.0–1.2) | 1.6 (1.4–2.2) |
| **ALT (U/L): Median (IQR)** | 9.0 (8.0–9.0) | 1259.0 (208.0–4968.0) | 15.0 (13.0–19.0) | 1172.0 (387.5–3724.0) |
| **Haemoglobin (g/L): Median (IQR)** | 124.0 (114.8–134.5) | 135.0 (121.0–143.0) | 133.0 (123.0–142.5) | 139.0 (114.0–148.8) |
| **Sodium (mmol/L): Median (IQR)** | 140.0 (138.0–141.3) | 137.0 (135.0–139.0) | 139.0 (137.0–140.0) | 139.0 (137.0–140.0) |
| **Bilirubin (µmol/L): Median (IQR)** | 9.0 (6.0–14.5) | 19.0 (13.0–34.0) | 8.5 (6.0–14.0) | 21.0 (13.0–33.0) |

One DILI sample was excluded from analysis in study A before unblinding the data due to a sample quality issue.

*ALP* alkaline phosphatase, *ALT* alanine transaminase, *DILI* drug-induced liver injury, *INR* International normalised ratio, *IQR* interquartile range, *NSAID* nonsteroidal anti-inflammatory drugs, *secs* seconds, *SSRI* selective serotonin reuptake inhibitors, *WBC* white blood cells.

of the relative bias and limits of agreement will depend on the K18 level and the overall summaries shown here should be interpreted cautiously.

## Discussion

In this study, we have developed a POC assay and demonstrated that the SERS-LFIA was able to identify patients with liver injury due to paracetamol. We propose that this in vitro diagnostic could be used for the stratification of patients by their liver injury risk at presentation to the hospital. This would allow for targeted treatment with the established antidote (NAC) within the optimal therapeutic window.

There are examples of SERS-LFIA tests being coupled with portable Raman readers. Hassanain et al. exploited the 'point-and-shoot' feature available on many portable Raman readers to interrogate the test line of an LFIA using a 3D-printed adaptor to line it up with the laser aperture[24]. This meant that a laser spot and multiple acquisitions were necessary to interrogate the full test line, increasing the time-to-result. To increase the area of the test line analysed, Tran et al. developed a portable Raman reader which incorporated line illumination and a

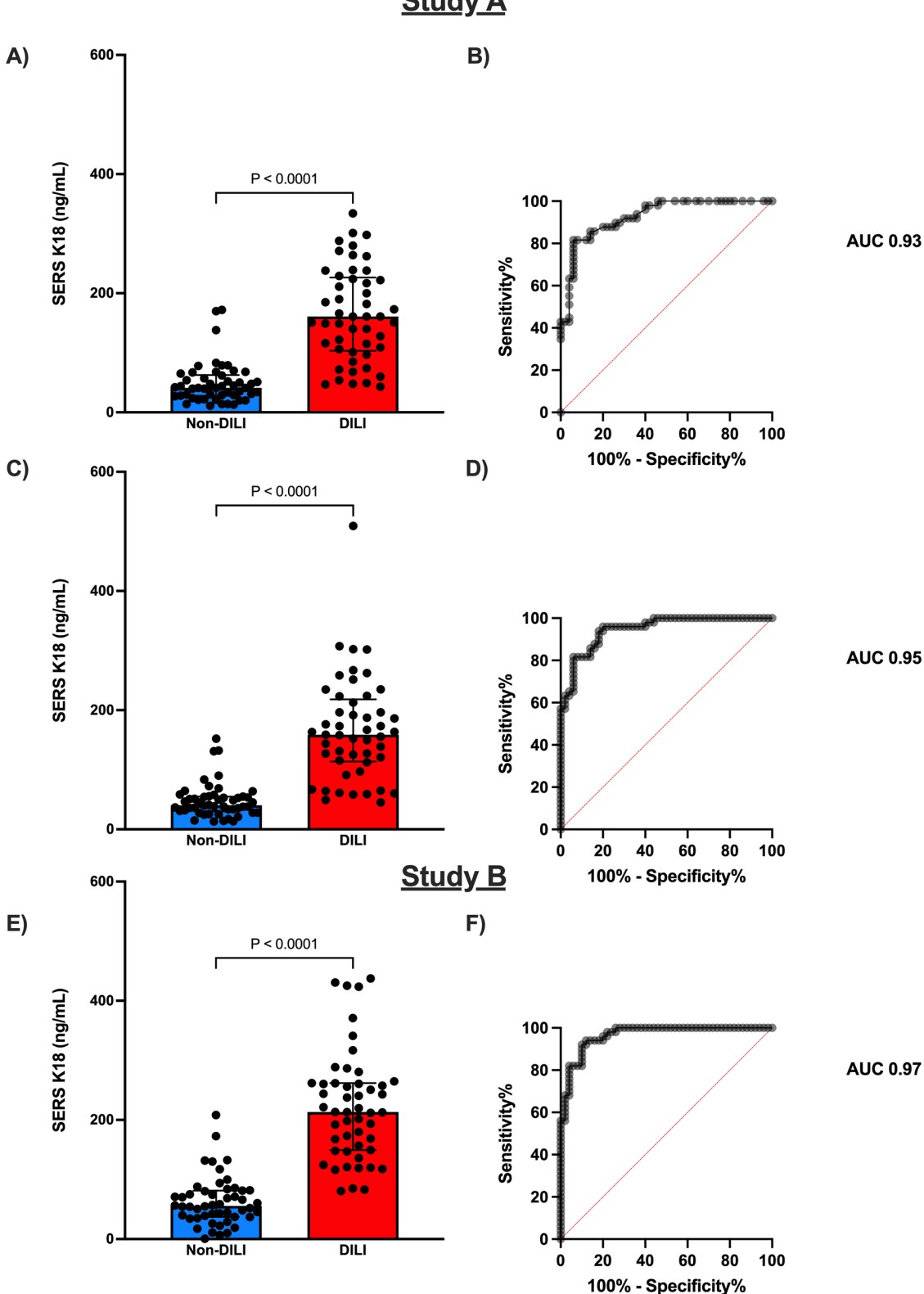

**Fig. 3 | Data for clinical studies A and B with serum samples from patients following a POD. A** Study A−Primary statistical output with randomly selected first values from the three analysers, with **B** corresponding ROC curve. **C** Study A−geometric mean data points for the three analysers with **D** corresponding ROC curve. **E** Refined version of the diagnostic evaluated in study B−mean data points for analyser with **F** corresponding ROC curve. Bars represent the median values ± interquartile range. *P* values are based on two-sided Mann−Whitney tests, *P* < 0.0001. AUC area under the curve, DILI drug-induced liver injury, K18 cytokeratin 18, ROC receiver operating characteristic, SERS surface-enhanced Raman scattering.

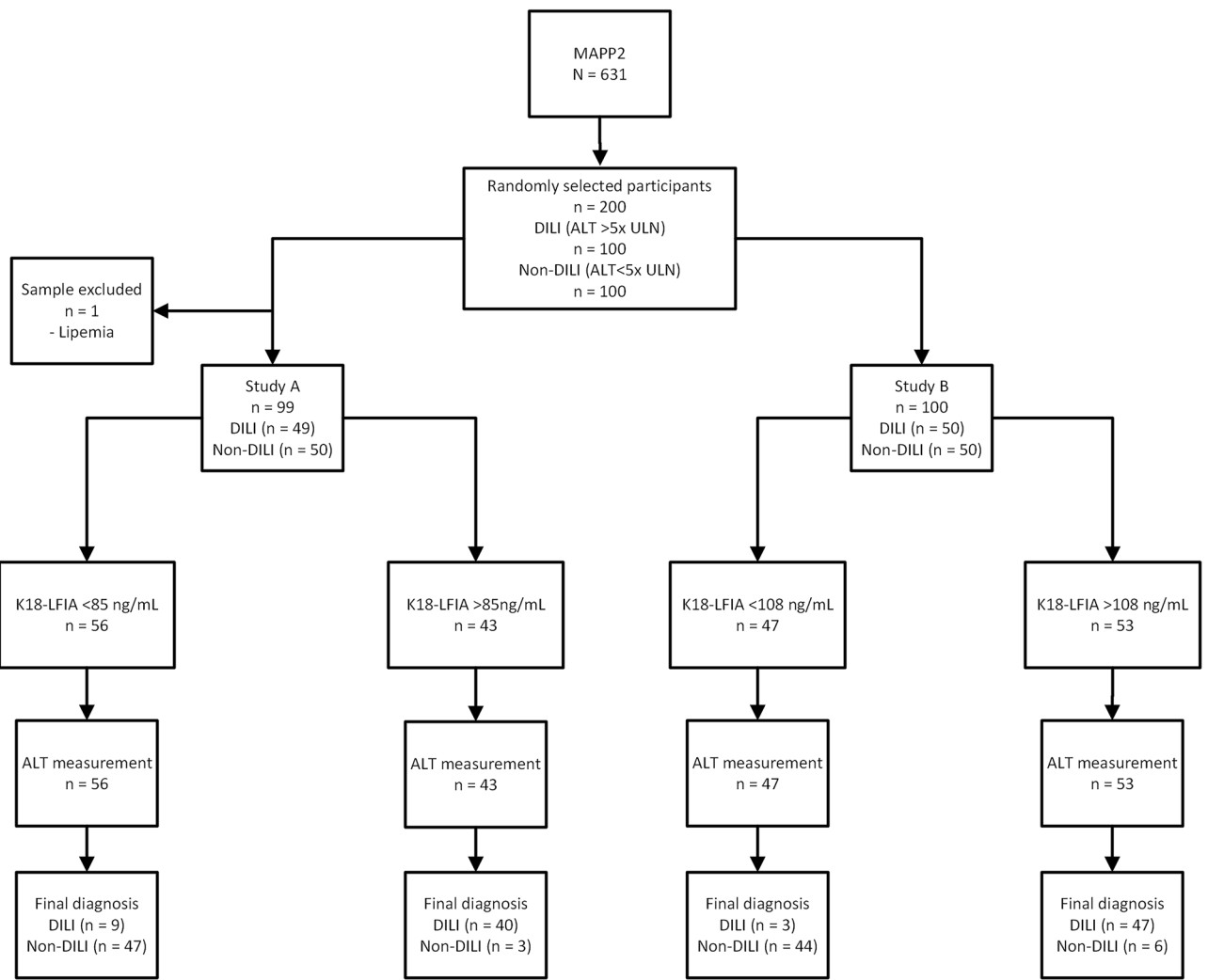

**Fig. 4 | STARD diagram to report the flow of participants through the study.** ALT alanine transaminase, DILI drug-induced liver injury, K18 cytokeratin 18, LFIA lateral flow immunoassay, MAPP2 Markers and Paracetamol Poisoning Study 2, ROC receiver operating characteristic, SERS surface-enhanced Raman scattering.

portable stage, which allowed the lateral flow strip to be moved through the laser, resulting in the whole width of the test line being analysed in 5 s[25]. The portable device was able to detect low concentrations of the pregnancy hormone human chorionic gonadotropin; however, the laser light was exposed, and this would not be safe for use in the Emergency Department. Joung et al. have developed a portable 'all-in-one' SERS-LFIA reader to diagnose SARS-CoV-2 onsite[26]. The portable reader contained a slot where the LFIA cassette was inserted, obscuring the laser light from the user, and the test and control lines were then Raman mapped using a 35 mW laser, taking 20 s to analyse 39 spots across the test line. By comparing their results to a commercial LFIA, they found that the sensitivity for 49 positive samples was 96%, significantly higher than the commercial LFIA's sensitivity of 57%. However, despite the excellent sensitivity, the portable instrument was still bulky and would need to be set up in a central lab, therefore limiting the POC scenarios it could be used in, especially where space is vital. Although these studies highlight the potential of SERS-LFIAs coupled to portable Raman readers for quantifying clinically relevant concentrations of biomarkers in a POC setting, they also emphasise developments which are still needed to produce a safe, rapid and truly portable Raman reader for SERS-LFIA that can be used in a clinical environment.

This study was performed using prototype SERS-LFIAs with the output measured using a bespoke HRR. In study A, the SERS-LFIA strips

were coupled to the HRR via a 3D-printed accessory, and in study B, sleeves were designed that held the SERS-LFIA cassette in the correct position. The change in coupling was made based on user feedback, which highlighted issues when aligning the strip with the laser. Adopting the second coupling method reduced the variability in lining up the test line with the laser, which was reflected in the increased sensitivity and specificity achieved in study B. The HRR reader used in Study B enclosed the laser within the reader so that laser light could not leak from the system, to improve safety during use in a clinical setting. This makes the HRR suitable for use in hospital Emergency Departments.

As expected, the visual analysis, by independent reviewers, had lower specificity than with SERS analysis and lacked the ability to provide a quantitative measure of K18, so it is purely qualitative. This is due to the subjective nature of visual analysis. Reviewers often interpreted negative tests as weakly positive visually, therefore impacting the false positive rate. The use of SERS measurements, which increased the specificity, produced quantitative K18 concentrations that are suitable for future prospective clinical studies.

The fundamental goal for developing a POC diagnostic is to achieve high sensitivity with a rapid time-to-result. The SERS-LFIA test provides a visual and quantitative result in under 30 min with high sensitivity. When compared with previous clinical studies[8,11,22,27–29], which evaluated K18 as a biomarker of DILI (paracetamol and non-

**Table 2 | Comparison of diagnostic accuracy between studies A and B**

| | Study A | | | | | Study B | | |
|---|---|---|---|---|---|---|---|---|
| | SERS analysis | | | | Visual analysis | SERS analysis | | Visual analysis |
| | Primary analysis Cut-off = 85 ng/mL | Primary analysis Cut-off = 49 ng/mL | Secondary analysis Cut-off = 91 ng/mL | Secondary analysis Cut-off = 58 ng/mL | | Cut-off = 132 ng/mL | Cut-off = 108 ng/mL | |
| Sensitivity (%) | 81.6 | 93.9 | 81.6 | 93.9 | 71.0 | 82.0 | 94.0 | 96.7 |
| Specificity (%) | 94.0 | 64.0 | 94.0 | 82.0 | 77.0 | 94.0 | 88.0 | 58.8 |
| Accuracy (%) | 87.9 | 78.8 | 87.9 | 86.9 | 73.0 | 88.0 | 91.0 | 77.9 |
| PPV | 5%—41.7 10%—60.2 20%—77.3 30%—85.4 | 5%—12.1 10%—22.5 20%—39.5 30%—52.8 | 5%—41.7 10%—60.2 20%—77.3 30%—85.4 | 5%—21.5 10%—36.7 20%—56.6 30%—69.1 | 5%—14.0 10%—25.5 20%—43.6 30%—57.0 | 5%—41.8 10%—60.3 20%—77.4 30%—85.4 | 5%—29.2 10%—46.5 20%—66.2 30%—77.0 | 5%—11.0 10%—20.7 20%—37.0 30%—50.1 |
| NPV | 5%—99.0 10%—97.9 20%—95.3 30%—92.3 | 5%—99.5 10%—99.0 20%—97.7 30%—96.1 | 5%—99.0 10%—97.9 20%—95.3 30%—92.3 | 5%—99.6 10%—99.2 20%—98.2 30%—96.9 | 5%—98.1 10%—96.0 20%—91.4 30%—86.1 | 5%—99.0 10%—97.9 20%—95.4 30%—92.4 | 5%—99.6 10%—99.2 20%—98.3 30%—97.2 | 5%—99.7 10%—99.4 20%—98.6 30%—97.6 |
| Positive LR | 13.6 | 2.6 | 13.6 | 5.2 | 3.1 | 13.7 | 7.8 | 2.3 |
| Negative LR | 0.2 | 0.1 | 0.2 | 0.1 | 0.4 | 0.2 | 0.1 | 0.1 |

SERS analysis using the primary analysis (randomly selected first SERS measurement from one of the three analysers) with cut-points that have the highest combination of specificity and sensitivity from the cut-points with close to 95% specificity or sensitivity. Secondary analysis with the geometric mean values is also presented. The visual analysis was performed by independent analysers who were blinded to the clinical classifications. PPV and NPV are presented with a range of DILI frequency in the population. PPV and NPV are presented with a range of DILI frequency in the population. *LR* likelihood ratio, *NPV* negative predictive value, *PPV* positive predictive value, *SERS* surface-enhanced Raman scattering.

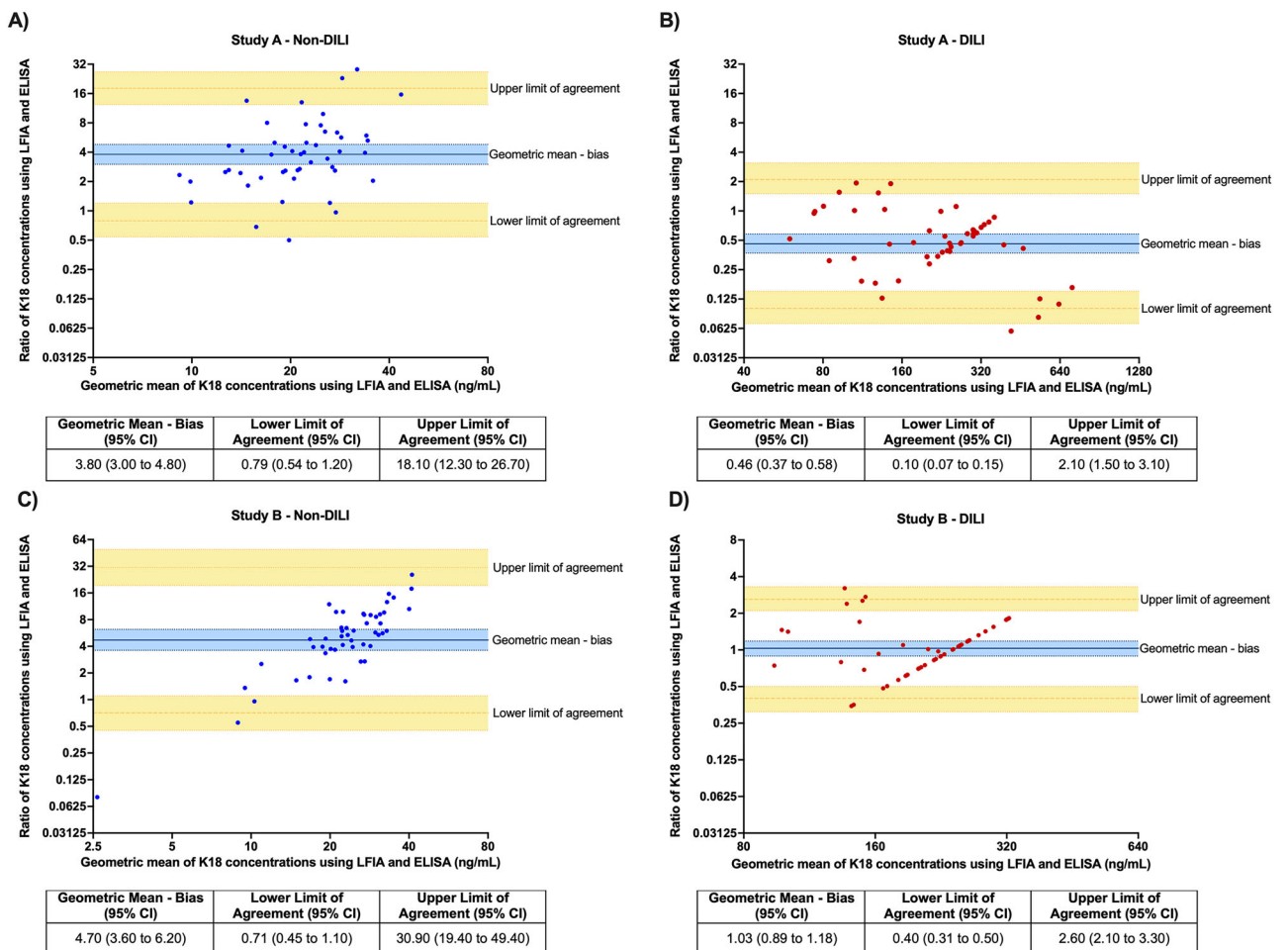

**Fig. 5 | Bland-Altman plots for log-transformed K18 concentration using SERS-LFIA and M65 ELISA.** Study A plots for **A** non-DILI, **B** DILI. Study B plots for **C** non-DILI, **D** DILI. The yellow horizontal lines represent the upper and lower limits of agreement (with 95% CI), and the blue line represents the geometric mean–bias.

DILI drug-induced liver injury, ELISA enzyme-linked immunosorbent assay, K18 cytokeratin 18, LFIA lateral flow immunoassay, SERS surface-enhanced Raman scattering.

paracetamol related) using the commercial ELISA, both of our studies produced comparable, or higher, ROC-AUC values (Supplementary Table 3). The SERS-LFIA can quantify the patient's circulating K18 concentration in less time than via an ELISA, or ALT concentration, has a high level of diagnostic accuracy, and the test can be carried out at the POC rather than a centralised laboratory.

This evaluation study of the SERS-LFIA test has demonstrated that SERS is an appropriate tool for the rapid detection of DILI following a POD. This study is an important development for POC diagnostics for use in an emergency setting, where treatment delays can directly impact patient outcomes. Accurate and rapid detection of DILI is essential for triaging and targeting therapeutic intervention to the patients most at risk of developing liver injury. Lateral flows produce rapid results, are low-cost and require minimal training to use, making them suitable for an emergency setting. The LFIA, coupled with the bespoke HRR, proved to be a robust, sensitive, specific and accurate tool for identifying patients at risk of DILI following a POD. The results of this study suggest that the diagnostic could offer an alternative to, or complement, the gold standard liver function tests, including ALT, and facilitate earlier intervention with antidote treatment. New therapeutic interventions are being developed for the treatment of POD[30,31]. Patient selection for clinical trials of these interventions could be enriched by the use of a POC test with the performance characteristics described in this paper (rule-in of liver injury). The rapid POC nature of the test will also allow for near-patient monitoring after administration of novel therapeutics. In clinical practice, early rapid identification of patients at risk of ALF would allow the implementation of new treatment pathways, which are particularly important given that NAC treatment efficacy is time-dependent. Conversely, with development, the SERS-LFIA test could also allow patients to be discharged sooner from the hospital once they are deemed to be low risk for DILI (rule-out of liver injury). This has the potential to reduce the bed occupancy of Emergency Departments and may be cost-saving. To achieve these objectives, the next step in the product development pathway is a prospective performance evaluation in capillary blood. It is important in future development that patients from diverse ethnic backgrounds are included, given that the data reported in this paper are predominantly from a white Scottish population.

Defining an optimal cut-off value for the SERS-LFIA using capillary blood is crucial to ensure accurate detection of DILI in a real-time clinical context. This study provides an initial reference point in serum; however, capillary blood has distinct physiological characteristics that may influence biomarker concentrations and test performance. Carefully determining the cut-off value through prospective cohort analysis would help to mitigate potential variations in analyte levels, minimise misclassification errors, and maintain the test's diagnostic sensitivity and specificity when transitioning to a more convenient and patient-friendly sampling method like capillary blood.

For both studies, there is good agreement between the K18 concentration obtained from the SERS-LFIA and ELISA tests for patients with DILI, with both producing high K18 concentrations. However, in non-DILI patients, the SERS-LFIA test reported higher K18 concentrations compared to the ELISA. We hypothesise that this occurred due to differences in assay protocols[32]. The ELISA involves multiple wash steps designed to remove non-specific binding, resulting in consistently low K18 concentrations in non-DILI samples. In contrast, the SERS-LFIA lacks these wash steps to enable rapid testing, which may allow non-specific proteins to remain and contribute to the overall signal. An additional factor that could contribute to the elevated K18 signal in non-DILI samples is the high density of antibodies on the nanoparticle surface used in the SERS-LFIA format. As noted by Sotnikov et al., steric restrictions associated with densely immobilised antibodies can enhance antigen binding in the context of protein targets such as K18[33]. This may account for the greater antigen-antibody interactions observed in the SERS-LFIA compared to ELISA, particularly at lower K18 concentrations. Finally, the calibration curve was created by spiking K18 protein into healthy serum, which will result in over-estimation of K18 concentration at the lower range due to endogenous K18 in the healthy serum.

In conclusion, the combination of lateral flow technology with an HRR allowed the sensitive and specific detection of DILI at the point of care with a rapid turnaround time. With development, this technology promises to cost-effectively improve patient outcomes. As it represents a platform technology, there is potential for measurement of other biomarkers either in isolation or multiplexed on a single assay.

## Methods

### Ethical statement
The serum samples assessed in studies A and B were randomly selected from the MAPP2 (ClinicalTrials.gov identifier: NCT03497104). Ethical approval for this study was provided by London−South East Research Ethics Committee (18/LO/0894), and all patients provided written informed consent. The study established a biobank of human serum samples from patients with POD. The serum was used to develop and test the performance of the SERS-LFIA. MAPP2 was a prospective, observational cohort study of participants presenting to the Emergency Department at the Royal Infirmary, Edinburgh. Inclusion criteria were that participants must be age 16 years and over, attending a hospital with a POD alone or as part of a mixed overdose, and able to give written informed consent.

### Materials
Sodium tetrachloroaurate(iii) dihydrate ($NaAuCl_4–H_2O$), sodium citrate, 4,4'-dipyridyl (DIPY), 3-(aminopropyl) trimethoxysilane (APTMS), sodium silicate, boric acid, borate, sodium chloride, Tween 20 and sucrose were all purchased from Merck (UK). PBS tablets were purchased from Oxoid. Recombinant human cytokeratin 18 protein, recombinant anti-cytokeratin 18 antibody (capture and detection) and goat anti-rabbit IgG were all purchased from Abcam (USA). Whatman MF1 bound glass fibre filter (conjugate pad), Whatman standard 14 (sample pad), Whatman CF6 dipstick pad (absorbent pad) and Nitrocellulose membrane FF170HP were purchased from Cytiva (USA). Polylactic acid (PLA) 3D printer filaments were purchased from Farnell (UK).

### Instruments
Gold nanoparticles were characterised using an Agilent Cary 60 UV-Vis spectrophotometer, Malvern Zetasizer Nano ZS and CBEx handheld Raman spectrometer with 785 nm laser excitation. Antibody test and control lines were added to nitrocellulose using a Claremont Bio Automated lateral flow reagent dispenser and at Lateral DX using an Imagene Technology lateral flow reagent dispenser. A Biodot CM500 Guillotine was used to cut nitrocellulose sheets into 5 mm strips, which were laminated using a Crenova Laminator. The cassettes were 3D printed using an Ultimaker S5. The test and control lines were measured using an HRR Wasatch Photonics with 785 nm laser excitation equipped with a 3D-printed LFIA cassette accessory (HRR 3A) or built-in slot (HRR 4A). For each HRR, the laser was a stabilised 785 nm laser diode from Innovative Photonic Solution rated for 100 mW, single mode. The laser output was limited in hardware to <5 mW. Laser attenuation was achieved by limiting the drive current to the laser, ensuring the amount of power from the spectrometer emitted does not exceed the class 3R limit (3.6 mW) at the sample. The optical parameters of the HRR were 600–1800 cm$^{-1}$ range with a spectral range of <15 cm$^{-1}$. A Powell lens was used to produce a stripe illumination ~3 × 1 mm. We used an integrated laser and probe whereby the laser line illumination and collection image are colinear with the collection optics having an NA of f/1.1.

## Experimental

**Gold nanoparticle synthesis.** All glassware was cleaned using Aqua regia and rinsed prior to use. Gold nanoparticles (AuNP) were synthesised using a refined Turkevich synthesis[34,35] by adding $NaAuCl_4$–$H_2O$ (67.5 mg) to distilled water (500 mL) in a 1 L, 3-necked round-bottom flask. The solution was heated for 30 min followed by the addition of sodium citrate (60.5 mg) and heated for an additional 15 min before being cooled to room temperature. Constant stirring was maintained throughout using a glass-linked stirrer. The solution was left to cool whilst being stirred overnight and then characterised using extinction spectroscopy and dynamic light scattering (DLS).

### Raman reporter functionalisation and silica encapsulation

AuNP (100 mL) was added to a 250 mL conical flask with a stirrer bar inside. DIPY (300 μL, 100 μM) was added to the solution, which was stirred for 5 min. APTMS (150 μL, 3 mM) and sodium silicate (1.5 mL) were then added. The solution was heated to 98 °C with constant stirring for 30 min and left to cool whilst being stirred overnight. The resulting Au-DIPY-$SiO_2$ NP were characterised using extinction spectroscopy, DLS and solution SERS using a 785 nm laser excitation CBEx spectrometer.

### Antibody functionalisation

Borate buffer (10 mM, pH 9, 100 μL) was added to Au-DIPY-$SiO_2$ NP (1 mL) in a low-binding Eppendorf tube. Recombinant anti-cytokeratin 18 antibody (4 μL, 1 mg/mL-study A, or 2 μL, 1 mg/mL, study B) was added to the solution and left to shake for 2 h. BSA (100 μL, 1% solution) was then added and the solution left to shake for an additional 30 min. The resulting Au-DIPY-$SiO_2$-Ab NP were characterised using extinction spectroscopy, DLS and SERS using a 785 nm excitation laser CBEx spectrometer. The Au-DIPY-$SiO_2$-Ab NP were then centrifuged at $2000 \times g$ for 20 min, the supernatant removed and the pellet resuspended in 100 μL of double-distilled deionised water. They were stored at 5 °C until required.

### Lateral flow strip preparation

**Lab-based line addition—study A.** Capture recombinant anti-cytokeratin 18 antibody (1 mg/mL) and goat anti-rabbit IgG (1 mg/mL) were added to the nitrocellulose section of the lateral flow sheet using a Claremont reagent dispenser flow reagent dispenser at a rate of 0.093 μL mm$^{-1}$ and left to dry overnight (room temperature, 16–24 h).

**Industry-based line addition—study B.** Capture recombinant anti-cytokeratin 18 antibody (1 mg/mL) and goat anti-rabbit IgG (1 mg/mL) were added to the nitrocellulose section of the lateral flow sheet using an Imagene Technology lateral flow reagent dispenser using a 300 mm dispense distance, 30 μL/S aspirate rate, 0.1 μL/mm dispense rate and 15 mm/s speed. The sheets were left to dry in an oven overnight (37 °C, 16–24 h).

### Pad preparation

Sample pad 1 (30 × 1 cm) was treated with NaCl (1% in distilled water), sample pad 2 (30 × 1.5 cm) was treated with Tween 20 (5% in water) and the conjugate pad (30 × 1 cm) was treated with sucrose and Tween 20 (1% and 0.5% in distilled water). The pads were then dried in an oven (37 °C, 3 h). The treated conjugate pads were then cut into 5 mm strips and the concentrated Au-DIPY-$SiO_2$-Ab NPs solution (10 μL, study A and 8 μL, study B) was pipetted onto the pad. The pads were dried in the oven (37 °C, 3 h).

### 3D-printed cassette

Cassettes that hold the lateral flow strip were 3D printed on an Ultimaker S5 with a 0.4 mm extrusion point using black tough PLA filament.

### Preparation and assembly

To assemble the LFIA, tape 1 on the antibody-treated nitrocellulose plastic-backed sheet was removed first. The 30 × 2 cm absorbent pad was added to this section. Tapes 2 and 3 were then removed. Sample pad 2 was added 0.5 cm up from the bottom of the strip. Sample pad 1 was added to the bottom of the strip with an overlap between pads 1 and 2. The sheet was laminated and left overnight to ensure adherence between the pads and sheet. The sheet was then cut into 5 mm strips using a biodot guillotine. The 5 mm conjugate pad loaded with Au-DIPY-$SiO_2$-Ab NP was added underneath the 2nd sample pad, and the strip was laminated again.

The strip was then placed into the PLA 3D-printed cassette. In study A, the devices were stored in a dark cupboard with no extra precautions. In study B, the devices were placed in a sealed foil bag containing a silica packet, then stored in a dark cupboard.

### Calibration curve—study A

A calibration curve developed for study A relating K18 concentration and SERS output was performed using the SERS-LFIA. Samples were prepared by mixing healthy human serum (25 μL) with buffer (75 μL, 5% Tween 20 in 10 mM PBS), which was spiked with 0, 5, 10, 25, 50, 100, 200, 350, 500 or 750 ng/mL of recombinant human cytokeratin 18 protein. The samples were then added to the sample port of a device and left to run for 20 min before analysis. The calibration was performed in healthy human serum from 3 donors to build a calibration curve containing averages and standard deviations.

### Calibration curve—study B

A calibration curve developed for study B relating K18 concentration and SERS output was performed using the SERS-LFIA. Samples were prepared by mixing healthy human serum (25 μL) with buffer (75 μL, 5% Tween 20 in 10 mM PBS), which was spiked with 0, 5, 10, 25, 50, 100, 200 ng/mL of recombinant human cytokeratin 18 protein. The samples were then added to the sample port of a device and left to run for 20 min before analysis. The calibration was performed in healthy human serum from 3 donors to build a calibration curve containing averages and standard deviations.

### SERS analysis

The test and control lines on the LFIA were analysed using an HRR with Enlighten 4.0.11 software. In study A, the LFIA cassette was slid into the adaptor, and the test and control area were lined up with the laser line (HRR 3A). Initial data were assessed using Microsoft Excel. In study B, the LFIA was analysed using an HRR 4A unit. The LFIA was put into cassette adaptors and inserted into the slot within the unit. All lines were analysed using a 785 nm laser excitation, 3.5 mW laser power, 1 s accumulation with 5 averages. To determine the signal ratio between the test line and the control line, linear regression was used using the respective intensities of the two spectra at corresponding pixels. The analysis was limited to the spectral range 750–1750 cm$^{-1}$, all spectra were baseline-corrected using air PLS in second order with lambda = 1e$^4$. No additional pre-processing was applied.

### Patient recruitment

Serum samples were randomly selected from the MAPP2 (ClinicalTrials.gov identifier: NCT03497104). Surplus blood samples were taken and stored at −80 °C. DILI was defined as an ALT elevation of ≥5 times the ULN. A schematic of the flow of study participants is presented in Fig. 4. Study demographics are detailed in Table 1, including self-reported gender.

The results obtained during the study did not inform any clinical decisions or patient management.

## ALT assay

ALT was measured in human serum by the NHS Lothian Biochemistry Laboratory. The ALT automated assay was performed using a Cobas C 701/702 system (Roche) with an Alanine Aminotransferase acc. to IFCC without pyridoxal phosphate activation test (Roche), as per the manufacturer's guidelines.

## K18 ELISA

The M65 ELISA was conducted at room temperature ($20 \pm 3\,°C$). Reagents were prepared following the manufacturer's guidelines. Standards, high and low controls and samples with unknown concentrations (25 mL each) were added to the microplate in duplicate. To each well, 75 µL of diluted M65 HRP Conjugate solution was added sequentially within 20 min. The microplate was sealed to prevent evaporation and contamination, then incubated on a shaker for 2 h at 600 rpm to facilitate the binding of the M65 HRP Conjugate to the target analyte. Following incubation, the plate was washed five times with 400 µL of wash solution per well using a plate washer. Subsequently, 200 µL of TMB Substrate was added to each well. The plate was incubated in the dark at room temperature for $20 \pm 1$ min, allowing the enzymatic reaction to take place. To stop the reaction, 50 µL of Stop Solution was added to each well, and the microplate was gently shaken for 10 s to ensure thorough mixing. Absorbance was measured after 5 min at 450 nm. The recorded absorbance values served as indicators of the analyte concentrations in the samples.

To ensure quality control, each ELISA run underwent specific test procedures. Firstly, the provided standards were assessed to ensure a simple linear regression with an appropriate $R$-squared value, confirming the reliability of the standard curve. Secondly, all samples were expected to yield values that could be interpolated from the standard curve. In case this criterion was not met, samples were re-measured at a 1:1 dilution with Standard A to ensure accurate quantification of the analyte.

## Clinical study A

A blinded, clinical study of 100 patient samples was performed to assess the sensitivity and specificity of the SERS-LFIA. The serum samples consisted of 50 with and 50 without DILI (sample size as per Clinical and Laboratory Standards Institute guidelines for assay development)[36], which was determined by measuring ALT concentration at presentation. Study data were collected and managed using REDCap (Research Electronic Data Capture) tools hosted at the University of Edinburgh[37,38]. Patient clinical information and reference standard results, including ALT and K18 ELISA data, were uploaded prior to study commencement to a REDCap database by an independent investigator and access was restricted for all analysers for the duration of the study.

Serum samples (25 µL) were thawed and added to buffer (5% Tween 20 in 10 mM PBS, 75 µL). This sample was then applied to the sample port of the cassette containing the LFIA strip. The assay was allowed to run for 20 min, and then the test and control lines were analysed using the HRR 3A. Each sample was run in triplicate and analysed three times by three individual analysers. To determine the concentration of K18 in the patient sample, the SERS of the test and control lines were analysed using linear regression to obtain the slope, which was then plotted on the calibration curve to generate a K18 concentration, performed using R Studio. One DILI sample was excluded from analysis before unblinding, due to visible lipemia.

The K18 concentration, determined by the SERS-LFIA, was input by the three individual analysers to a REDCap database and locked before analysis was performed by the Edinburgh Clinical Trials Unit.

## Clinical study B

A further blinded, clinical study of 100 patient samples was performed to assess the sensitivity and specificity of the SERS-LFIA, consisting of 50 samples with and 50 without DILI, which was determined by measuring the concentration of ALT at presentation. The samples were run on the SERS-LFIA, as described for study A, by a single analyser and analysed using the HRR 4A. The K18 concentration (determined by SERS analysis) was uploaded for each patient sample and compared to reference standard results by an independent investigator.

## Statistics and reproducibility

The study was evaluated using a pre-defined statistical analysis plan (SAP). Statistical analysis for study A was performed and reported using SAS Software 9.4 and Stata 16, after the REDCap database was locked. The primary analyses for the performance of the SERS-LFIA were performed with one randomly selected measurement (rounded to the nearest integer) from the first K18 measurements of each of the three analysers. The secondary analyses assessed the K18 measurement obtained by each individual analyser and geometric mean values for the three analysers.

Sample sizes were pre-defined in the SAP and as per the CLSI EP12-A2 guideline for assay development to enable the sensitivity/specificity to be estimated, using a Normal approximation to the binomial distribution, with confidence interval width ±0.099 (for a true value of 0.85) and ±0.071 (for a true value of 0.93).

Diagnostic accuracy of the SERS-LFIA was quantified via area under the ROC curve; sensitivity at the K18 cut-point with the highest combination of sensitivity and specificity among the cut-points with close to 95% specificity; and specificity, at a cut-point similarly defined. For all measures of accuracy, 95% confidence intervals (calculated via the Wilson method) were reported.

Positive and negative predictive values across a range of clinically plausible DILI prevalence rates [5%, 10%, 20%, 30%] were calculated for each of these prevalence rates and the sensitivity and specificity at the K18 cut-point which has highest combination of sensitivity and specificity from the cut-points with close to 95% specificity.

Statistical analysis for study B was performed using Prism 10.0 (GraphPad). The assay's accuracy, sensitivity, specificity, positive predictive value (PPV) and negative predictive value (NPV) were calculated with Prism 10.0.

All assay data points for study A and B were above the lower limit of quantification. In study A, one DILI sample was excluded from analysis before unblinding, due to visible lipemia. For studies A and B, the investigators were blinded to group classification during experiments and outcome assessment. The trial is reported as per the STROBE statement and STARD guidelines[39] (Supplementary Table 4) with extended details provided in the Supplemental Information.

## Reporting summary

Further information on research design is available in the Nature Portfolio Reporting Summary linked to this article.

## Data availability

The data generated in this study are provided in the Supplementary Information/Source Data file. The raw SERS data used in this study are available on the Pure portal (https://doi.org/10.15129/8fe6d3bc-25e3-4d18-aab1-b5a42c7b6645). Source data are provided with this paper.

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

## Acknowledgements

The study was funded by the Medical Research Council (MRC) DPFS Scheme. *MICA: Point-of-care assessment of drug-induced liver injury (POC DILI)* MRC Reference: MR/V038303/1 (S.S.D., K.M.S., B.C., P.F., J.M., S.L., C.J.W., J.W.D., K.F., D.G.). B.C. was supported by an EPSRC DTP studentship (EP/T517938/1) and Wasatch Photonics. K.M.S. was supported by The Centre for Precision Cell Therapy for the Liver, which was funded by the Chief Scientist Office (CSO) of the Scottish Government Health Directorates [PMAS/21/07]. The Research Electronic Data Capture (REDCap) system was used for data management. Edinburgh Clinical Trials Unit—Study A database was developed by Michelle Steven, with data management undertaken by Lynsey Milne. Study A statistical analysis was performed by Robert Lee.

## Author contributions

S.S.D., K.M.S. and B.C. performed the experiments and analysed the data. P.F., J.M. and S.L. project managed the programme of work. N.C.S., C.R., D.C., D.B., J.F. and M.Z. developed the handheld readers. E.V. and C.J.W. performed the statistical analysis. J.W.D., K.F. and D.G. supervised the project.

## Competing interests

C.R., D.C., D.B., J.F. and M.Z. are employees of Wasatch Photonics. S.S.D., K.M.S., B.C., J.W.D., K.F. and D.G. have filed a patent application related to the LFIA (Patent Application No. GB2314603.8, filed 22 September 2023). This technology relates to the detection methods described in the current study. The remaining authors declare no competing interests.
