## [Transparent Peer Review file · Nature Communications]

A new point-of-care diagnostic for drug-induced liver injury using surface-enhanced Raman scattering lateral flow immunoassay

Corresponding Author: Professor James Dear

Version 0:

Reviewer comments:

Reviewer #1

(Remarks to the Author)

The group from Edinburgh report on the development of a POC instrument to quantify serum Cytokeratin 18 levels from patients who had been enrolled in prior clinical trials of APAP overdose presenting to the ER. The rationale for the study and methods are well described and justified.

1. Data analysis- Two cohorts of patients in Study A and B are reported that had biobanked serum samples tested in the initial and modified Raman LFIA devices. The authors state that the sera of 50 patients with DILI (elevated ALT) and no DILI (ALT < ULN) at presentation were randomly selected for testing. Since the readout of the device is quantitative it would be of interest in the NO DILI patients to see if the quantitative level of CK-18 in those who went on to develop DILI during the study were higher compared to the non-DILI patients who never developed DILI. Based upon Ref #9 it is reasonable to speculate that subclinical DILI may be detected prior to ALT increase with this assay and it would be interest to know if the initial non-DILI cases who did develop DILI had elevated CK18 from the get go.

2. Abstract & Intro- In abstract, authors state that there 100,000 cases of APAP overdose in UK and in intro they state 50,000 cases. Please address and clarify.

3. Need for further studies- In the DISCUSSION, the need to define an optimal cut off for DILI vs non-DILI in a prospective cohort with capillary blood sampling and in a real time clinical context should be further emphasized.

4. Factors impacting CK-19 levels in health volunteers- The authors state that there was substantial intra and interindividual variability in the control and test bands of the LFIA. What factors are likely to contribute to this? Does total bilirubin levels impact performance characteristics of the test or lipid profile?

Reviewer #2

(Remarks to the Author)

Manuscript represents high quality research on rapid and accurate detection of drug-induced liver injury using Lateral Flow Immunoassay combined with handheld Raman Reader and surface-enhanced Raman scattering. Presented results on diagnostic accuracy for drug-induced liver injury with a specificity of 94% and sensitivity of 82% are well supported by satisfactory amount of clinical results for patients for this proof of concept study.

Nevertheless, presented method of SERS-LFIA is not novel as state of the art was well described in Discussion section. However, the application of drug-induced liver injury using SERS-LFIA is novel and very relevant for the development of point of care diagnostics.

Therefore, I suggest publication of this manuscript in Nature Communications if authors will reflect on my comments:

1. It has been mentioned in the manuscript that novel handheld Raman reader was developed (line 394). Therefore, I would expect to see the optical schema of the novel Raman reader with detailed description of all optical elements for laser beam delivery and Raman beam delivery. Please try to add supplementary Figure with detailed optical schema and detailed description of all elements and explanation of novelty of such custom spectrometer. In particular:

a. Please describe optical parameters of novel Raman reader: spectral range, resolution, throughput from sample to detector.

b. What laser was used: multimode or single mode? Max laser power measured just from the laser and on the sample? How

laser attenuation was done? What was the laser manufacturer?

c. What was the length of the laser line on the sample focal plane? Was the laser line homogeneously illuminated?

Typically, Powell lens provides inhomogeneous illumination on the sample, especially on the edges of the line. Was it taken into account and post corrected by the software via line intensity normalization?

d. What was the sample probe (microscope objective) NA, focal length, model?

2. It is mentioned that Raman reader is Class 3R. Is it a problem to make it Class 1? Do you plan to implement Class 1 in the future? Please discuss this in the manuscript.

3. I could not find detailed information on spectra analysis routine. Please add details on chemometric methods being used, spectral range being used, spectra postprocessing details like background correction or smoothing. It is not enough to mention that "spectra baseline corrected on Wasatch Photonics software" (line 75 in supplementary). Please describe baseline correction method and settings of the method.

Reviewer #3

(Remarks to the Author)

A comprehensive SERS-based study on the rapid diagnosis of severe liver injury induced by a paracetamol overdose is presented. Specifically, quantitative SERS results for the biomarker cytokeratin 18 in numerous patient samples are obtained by using a lateral flow immunoassay in conjunction with a customised handheld Raman instrument.

This study will certainly attract the attention of readers from applied and clinical chemistry since it is well performed on a statistically relevant number of real-world samples. However, other groups have already reported similar SERS-LFIA-based clinical studies on other diseases such as COVID-19 (many studies!), scrub typhus, as well as infections caused by respiratory bacteria, e.g., *Streptococcus pneumoniae*, and respiratory Virus, e.g., influenza A. This is complemented by various environmental studies using the SERS-LFIA technology, e.g. DOI: 10.1021/acsnm.4c03280. Even the same technical approach using a similar handheld device in combination with a line Focus has been published recently (DOI: 10.1039/D4SD00056K).

A major claim made by the authors is the ability to quantify the concentration of the target protein by SERS-LFIA. The qualitative diagnostic capability of SERS-LFIA (non-DILI/DILI) is clearly demonstrated by their ROC curve in Fig. 3 on p. 11. Although the data in Fig. 2d on p. 8 of their manuscript compare the performance of both SERS-LFIA and ELISA as a gold standard, this statistical analysis is not sufficient. The quantification performance of the two techniques must be separately evaluated. In other words, a statistical approach is needed to clearly compare the differences between the two techniques, for example, by Passing-Bablok regression or Bland-Altman plots (see *Sensors & Actuators: B. Chemical* (2024) - Fig. 5d in DOI: 10.1016/j.snb.2024.136078 and also *ACS Nano* (2017) - Fig. 6 a&b in DOI: 10.1021/acsnano.7b01536.)

Overall, this submission lacks the necessary novelty for acceptance in this journal. Submission to a journal devoted on analytical or clinical chemistry or *Sensors* is recommended.

Reviewer #4

(Remarks to the Author)

Version 1:

Reviewer comments:

Reviewer #1

(Remarks to the Author)

The authors appear to have addressed my concerns and comments as well as those of the other reviewers.

Reviewer #2

(Remarks to the Author)

I fully accept all modifications regarding my comments, and I accept manuscript for publication.

We thank the reviewers for taking the time to provide us with valuable feedback that has helped us to improve our manuscript. We have addressed all the points raised and believe the paper has been substantially strengthened as a result. We hope you find these corrections satisfactory and look favourably upon our revised manuscript. We have shown all changes as follows:

Italic – reviewers' comment

Red text – response

Highlighted text - excerpts from the revised Manuscript or Supporting Information.

REVIEWER COMMENTS

Reviewer #1 (Remarks to the Author):

The group from Edinburgh report on the development of a POC instrument to quantify serum Cytokeratin 18 levels from patients who had been enrolled in prior clinical trials of APAP overdose presenting to the ER. The rationale for the study and methods are well described and justified.

1. Data analysis- Two cohorts of patients in Study A and B are reported that had biobanked serum samples tested in the initial and modified Raman LFIA devices. The authors state that the sera of 50 patients with DILI (elevated ALT) and no DILI (ALT < ULN) at presentation were randomly selected for testing. Since the readout of the device is quantitative it would be of interest in the NO DILI patients to see if the quantitative level of CK-18 in those who went on to develop DILI during the study were higher compared to the non-DILI patients who never developed DILI. Based upon Ref #9 it is reasonable to speculate that subclinical DILI may be detected prior to ALT increase with this assay and it would be interest to know if the initial non-DILI cases who did develop DILI had elevated CK18 from the get go.

We thank the reviewer for highlighting this interesting point regarding the outcomes of the patients in the trial. Of the 100 non-DILI participants across study A and B, only one had a subsequent rise in ALT (study sample ALT = 8 U/L, peak ALT = 542 U/L (ULN=50)). The participant's ALT was not raised until 30 hours after the first sample (analysed in study B) was taken. In the study sample the K18 was in the non-DILI range (SERS K18 = 43.0 ng/mL - median non-DILI SERS K18 = 55.5 ng/mL, M65 ELISA = 16.0 ng/mL – median non-DILI M65 ELISA = 10.0ng/mL). This may reflect K18 having not yet increased because the study sample was taken only 4 hours after ingestion of the paracetamol overdose.

Given that there was only one patient, we have not added these data to the main manuscript. We are happy to add this if the Editor feels that it would be beneficial for the readership.

2. *Abstract & Intro- In abstract, authors state that there 100,000 cases of APAP overdose in UK and in intro they state 50,000 cases. Please address and clarify.*

We thank the reviewer for this comment. We have rephrased this in the Introduction as follows
“Paracetamol overdose (POD) is common, with around 100,000 people presenting to emergency departments following a POD and approximately 50,000 patients requiring emergency antidote treatment to prevent drug-induced liver injury (DILI), and subsequent ALF, every year in the UK alone.⁴”

3. *Need for further studies- In the DISCUSSION, the need to define an optimal cut off for DILI vs non-DILI in a prospective cohort with capillary blood sampling and in a real time clinical context should be further emphasized.*

Thank you, this is an important point. We are performing a prospective diagnostic performance evaluation in capillary blood at the time of writing (<https://www.isrctn.com/ISRCTN11484200>).

We have revised the manuscript as per your suggestion. We have added the following text to the Discussion:

“Defining an optimal cut-off value for the SERS-LFIA using capillary blood is crucial to ensure accurate detection of DILI in a real-time clinical context. This study provides an initial reference point in serum; however, capillary blood has distinct physiological characteristics that may influence biomarker concentrations and test performance. Carefully determining the cut-off value through prospective cohort analysis would help to mitigate potential variations in analyte levels, minimise misclassification errors, and maintain the tests diagnostic sensitivity and specificity when transitioning to a more convenient and patient-friendly sampling method like capillary blood.”

4. *Factors impacting CK-19 levels in health volunteers- The authors state that there was substantial intra and interindividual variability in the control and test bands of the LFIA. What factors are likely to contribute to this? Does total bilirubin levels impact performance characteristics of the test or lipid profile?*

As the reviewer points out, there was intra and inter-individual variability in the control and test band measurements of the LFIA in healthy volunteers. In the SI of the manuscript, we have hypothesised that this is due to the 'serum effect'. We have expanded on this term in the SI to describe the factors that are likely to contribute to this and how we have mitigated them.

The following text has been added to the SI:

“Serum proteins can bind to the gold nanoparticle surface forming a protein corona with each serum sample producing its own unique corona. It is termed unique as volunteers have slightly different serum composition due to factors which influence protein production. This includes genetics, diet, health status and environmental impacts. The formation of the protein corona will impact the antibody-antigen interaction and in most cases, it reduces the non-specific binding¹. It can also impact the flow of the nanoparticles through the LFIA strip as larger particles will travel slower through the strip. As different coronas are formed, the binding will be slightly different between volunteers. This is reflected in the SERS signal with higher binding corresponding to a higher SERS signal. Fortunately, this effect has been minimised by normalising the binding of the test line to the binding of the control line using the SERS spectra.”

Reviewer #2 (Remarks to the Author):

Manuscript represents high quality research on rapid and accurate detection of drug-induced liver injury using Lateral Flow Immunoassay combined with handheld Raman Reader and surface-enhanced Raman scattering. Presented results on diagnostic accuracy for drug-induced liver injury with a specificity of 94% and sensitivity of 82% are well supported by satisfactory amount of clinical results for patients for this proof of concept study. Nevertheless, presented method of SERS-LFIA is not novel as state of the art was well described in Discussion section. However, the application of drug-induced liver injury using SERS-LFIA is novel and very relevant for the development of point of care diagnostics. Therefore, I suggest publication of this manuscript in Nature Communications if authors will reflect on my comments:

1. It has been mentioned in the manuscript that novel handheld Raman reader was developed (line 394). Therefore, I would expect to see the optical schema of the novel Raman reader with detailed description of all optical elements for laser beam delivery and Raman beam delivery. Please try to add supplementary Figure with detailed optical schema and detailed description of all elements and explanation of novelty of such custom spectrometer.

We thank the reviewer for their valuable input. We have added a figure showing the optical configuration of the Raman reader into the SI and added the following text to the manuscript to explain the optical set-up and the novelty of the custom spectrometer.

The following has been added to Results:

“Therefore, the novelty of this HRR lies in three elements: 1) use of a highly efficient f/1.1 transmissive design with a low-cost Complementary Metal-Oxide Semiconductor (CMOS) detector for enhanced sensitivity, 2) inclusion of a Powell lens to maximize overlap between the excitation laser and LFIA strip lines, with consequent direct imaging onto the detector, and 3) use of a Class 3R laser to effectively and repeatably perform SERS measurements on a lateral flow strip, thus reducing laser exposure risk to facilitate implementation in clinical settings. A detailed optical schema of the HRR is shown in Supplementary Figure 3.”

The following has been added to the Supplementary Information:

“Both handheld Raman readers (HRR 3A and 4A) are comprised of a 785 nm laser excitation source, spectrometer module with CMOS detector, and sampling optics to direct and image laser light into a line on the LFA strip, which also directs and filters the Raman scattered light for detection by the spectrometer. It also integrates an LFIA cassette holder accessory to house the LFIA cassette during signal acquisition, with two functions:

- 1) Locating and registering of the cassette for measurement of both control and sample lines, and
- 2) Enclosure of the cassette to mitigate possible exposure of the user to laser light.

The spectrometer module includes collimation and dispersion optics to spread incoming light by wavelength over a line array CMOS sensor. The diagram (Supplementary Figure 3) shows the optical path, isolated from the overall opto-electronics diagram. Light is emitted from the laser diode and passes through a Powell lens that images the Gaussian laser beam into a line of uniform intensity, which is then directed onto the sample by a dichroic long-pass mirror and focusing lens. The resulting Raman scattered light is collected by the same focusing lens, passes through the dichroic long-pass mirror and a long-pass dichroic filter to reject scattered laser light, and is directed through the slit focus assembly, a collimator assembly, and onto the diffraction grating. Light is diffracted through the grating, into the focusing lens assembly, and is directed onto the CMOS sensor for detection.”

Supplementary Figure 3. Diagram of optical path through the HRR to the LFIA line.

a. Please describe optical parameters of novel Raman reader: spectral range, resolution, throughput from sample to detector.

We thank the reviewer for bringing this omission to our attention. The HRR Raman readers had the following optical parameters; f/1.1 input, 600 - 1800 cm^{-1} range with $<15 \text{ cm}^{-1}$ average FWHM resolution. The throughput from sample to detector was not determined as it was based upon detection of sufficient signal.

This information has now been included in the Instrument section.

b. What laser was used: multimode or single mode? Max laser power measured just from the laser and on the sample? How laser attenuation was done? What was the laser manufacturer?

A stabilised 785 nm single mode laser diode from Innovative Photonic Solutions (IPS) was used which had a maximum power output of 100 mW. However, the laser power output was limited to $<5 \text{ mW}$ before mounting within the instrument and the laser power at the sample was 3.6 mW. The laser attenuation was achieved by limiting the drive current to the laser ensuring the amount of power the spectrometer emits does not exceed the class 3R limit ($<5 \text{ mW}$).

We thank the reviewer for bringing this to our attention and we have included additional information in the Instrument section.

c. What was the length of the laser line on the sample focal plane? Was the laser line homogeneously illuminated? Typically, Powell lens provides inhomogeneous illumination on the sample, especially on the edges of the line. Was it taken into account and post corrected by the software via line intensity normalization?

The Powell lens produced a strip of illumination approximately 3 mm x 1 mm in size, the goal was to overfill the LFA sample region of interest. We did not perform studies to determine if the laser line was homogeneously illuminated as sufficiently reproducible signals were obtained across the line compared to point based measurements. Therefore, no post-correction of line intensity normalization was carried out. In the CMOS detector we vertically bin pixels to produce a single intensity reading for each horizontal row of pixels. This ensures we not only get good overlap of the LFA but mitigate for any laser power distribution effects in the long axis of the sample.

Our focus was ensuring that we had reproducible measurements which was achieved by controlling the position of the lines using the cassette holder, which allowed the user to precisely position the LF line in front of the laser beam. This allowed repeatability of measurements between strips as well as users.

Additional information on the Powell lens has been added to the Instrument section.

d. What was the sample probe (microscope objective) NA, focal length, model?

Thank you for alerting us to this omission. We used an integrated laser and probe whereby the laser line illumination and collection image are colinear with the collection optics having an NA of f/1.1.

The complete text added to the Instrument section to address a, b, c, and d comments is as follows:

“For each HRR, the laser was a stabilised 785 nm laser diode from Innovative Photonic Solution (IPS) rated for 100 mW, single mode. The laser output was limited within the instrument to <5 mW. Laser attenuation was achieved by limiting the drive current to the laser, ensuring the amount of power the spectrometer emits does not exceed the class 3R limit (3.6 mW at the sample). The optical parameters of the HRR were 600 - 1800 cm^{-1} range with a spectral resolution of <15 cm^{-1} . A Powell lens was used to produce a stripe illumination

approximately 3 mm x 1 mm. We used an integrated laser and probe whereby the laser line illumination and collection image are colinear with the collection optics having an NA of f/1.1.”

2. *It is mentioned that Raman reader is Class 3R. Is it a problem to make it Class 1? Do you plan to implement Class 1 in the future? Please discuss this in the manuscript.*

Thank you for this question as it is a valuable point to highlight. The Raman reader effectively was a Class 1 device as the laser was fully enclosed within the device and sampling accessory. However, we did not carry out the certification process to confirm this so cannot definitively make this claim in the manuscript. This will be looked at in the future but was beyond the timescales of the current study.

To address this, the following has been added to the Results section of the manuscript:

“In practice, the interface between the laser and sample are fully enclosed in HRR 3A and 4A, effectively rendering the device as Class 1, as it meets the definition ‘Class 1 lasers have low radiated power or are enclosed to prevent radiation from escaping’. However, as we are without certification, we have classed them both as 3R.”

3. *I could not find detailed information on spectra analysis routine. Please add details on chemometric methods being used, spectral range being used, spectra postprocessing details like background correction or smoothing. It is not enough to mention that "spectra baseline corrected on Wasatch Photonics software" (line 75 in supplementary). Please describe baseline correction method and settings of the method.*

Thank you for bringing this detail to our attention. To address this, the following text has been added to the Methods section of the manuscript:

“To determine the signal ratio between the test line and the control line, linear regression was used using the respective intensity of the two spectra at corresponding pixels. The analysis was limited to the spectral range 750-1750 cm^{-1} , all spectra were baseline-corrected using air PLS in second order with $\lambda = 1e^4$. No additional pre-processing was applied.”

Reviewer #3 (Remarks to the Author):

A comprehensive SERS-based study on the rapid diagnosis of severe liver injury induced by a paracetamol overdose is presented. Specifically, quantitative SERS results for the biomarker

cytokeratin 18 in numerous patient samples are obtained by using a lateral flow immunoassay in conjunction with a customised handheld Raman instrument.

*This study will certainly attract the attention of readers from applied and clinical chemistry since it is well performed on a statistically relevant number of real-world samples. However, other groups have already reported similar SERS-LFIA-based clinical studies on other diseases such as COVID-19 (many studies!), scrub typhus, as well as infections caused by respiratory bacteria, e.g., *Streptococcus pneumoniae*, and respiratory Virus, e.g., influenza A. This is complemented by various environmental studies using the SERS-LFIA technology, e.g. DOI: 10.1021/acsanm.4c03280. Even the same technical approach using a similar handheld device in combination with a line Focus has been published recently (DOI: 10.1039/D4SD00056K).*

With respect to the Reviewer, our approach of combining a bespoke HRR, SERS measurements and LFIA platform is novel for the detection of DILI in serum. This is the first point-of-care DILI test designed to detect and quantify K18 levels, and it could lead to improved patient stratification if implemented in hospital emergency departments. Our robust study, which was performed with 200 serum samples, with matched numbers of positive and negative samples, and under clinical trial conditions is a much larger cohort tested than in previous papers. For example, the detection of scrub typhus via SERS-LFIA (DOI: 10.1021/acs.analchem.9b02363) used only 16 positive and 24 negative patient samples, a report on COVID-19 detection via SERS-LFIA (DOI: 10.1021/acssensors.2c01808) used 49 positive and 5 negative patient samples to evaluate the clinical effectiveness and the detection of influenza A (doi.org/10.1016/j.snb.2024.136078) used 21 positive and 10 negative patient samples to validate the clinical value. The detection of chloramphenicol residue with SERS-LFIA (DOI: 10.1021/acsanm.4c03280) did test a large number of environmental samples but the number of positive and negative samples were not equal (54 negatives and 148 positives). Having a large, equal number of positive and negative samples is crucial for ensuring the accuracy and reliability of the test. An imbalance in sample size can introduce bias, potentially overestimating or underestimating a tests performance.

Furthermore, the rapid assessment which provides a result 30 minutes after sample addition has also produced a clear pathway for how SERS-LFIA paired with a portable HRR could be implemented in hospital emergency departments. This rapid analysis and portable aspects

are often not mentioned in SERS-LFIA publications making it difficult to understand how SERS-LFIA can be translated into POC scenarios.

Although an early version of the HRR has been published, we have emphasised the importance in the pairing of the LFIA cassettes and HRR, and we have adapted the HRR 4A to focus on increasing the reproducibility between tests by implementing inserts designed to hold the LFIA cassette in the correct position without the need for manual moving. The reader also operates as a class 3R allowing measurements to be taken safely, another vital step when moving the HRR into POC environments. Overall, we believe that these changes have allowed the test to achieve high sensitivity and specificity when diagnosing DILI in real life samples.

A major claim made by the authors is the ability to quantify the concentration of the target protein by SERS-LFIA. The qualitative diagnostic capability of SERS-LFIA (non-DILI/DILI) is clearly demonstrated by their ROC curve in Fig. 3 on p. 11. Although the data in Fig. 2d on p. 8 of their manuscript compare the performance of both SERS-LFIA and ELISA as a gold standard, this statistical analysis is not sufficient. The quantification performance of the two techniques must be separately evaluated. In other words, a statistical approach is needed to clearly compare the differences between the two techniques, for example, by Passing-Bablok regression or Bland-Altman plots (see Sensors & Actuators: B. Chemical (2024) - Fig. 5d in DOI: 10.1016/j.snb.2024.136078 and also ACS Nano (2017) - Fig. 6 a&b in DOI: 10.1021/acsnano.7b01536.)

Overall, this submission lacks the necessary novelty for acceptance in this journal. Submission to a journal devoted on analytical or clinical chemistry or Sensors is recommended.

We thank the reviewer for their detailed feedback.

As the reviewer has suggested, we have evaluated the quantification performance of SERS-LFIA and ELISA by Bland-Altman analysis. The plots presented for the two studies for non-DILI and DILI participants (Figure 5). The analysis was performed by Elizabeth Varghese (Clinical Trials Statistician in Edinburgh Clinical Trials Unit) who has been added to the author list. The differences between the log transformed K18 measurements using SERS and ELISA have a reasonably normal distribution, but there is evidence of an association between the differences in the log transformed K18 measurements and the mean of the log transformed

K18 measurements using SERS and ELISA so the estimates of the relative bias and limits of agreement will depend on the K18 level and the overall summaries shown here should be interpreted cautiously.

We would like to highlight that the clinical need, and aim of our study, is to use the point-of-care test to discriminate between DILI and non-DILI cases. The assessment of agreement between SERS and ELISA was not a principal objective for the clinical study.

The following text has been added to the Results:

“Agreement for the quantification of K18 in the clinical samples was compared between SERS-LFIA and K18 (M65) ELISA. Bland-Altman plots are presented for study A and B in Figure 5. In both studies, the ratio between the two methods is lower for DILI than non-DILI (study A DILI = 0.5, non-DILI = 3.8 and study B DILI = 1.0 and non-DILI = 4.7). The non-DILI group also had a greater ratio between the upper and lower limits of agreement in comparison to DILI for both study A and B. The differences between the log transformed K18 measurements using SERS and ELISA have a reasonably normal distribution, but as there is evidence of an association between the difference in the log transformed K18 measurements and the mean of the log transformed K18 measurements using SERS and ELISA, the estimates of the relative bias and limits of agreement will depend on the K18 level and the overall summaries shown here should be interpreted cautiously.”

Figure 5. Bland-Altman plots for log transformed K18 concentration using SERS-LFIA and M65 ELISA. Study A plots for A) non-DILI, B) DILI. Study B plots for C) non-DILI, D) DILI. The yellow horizontal lines represent the upper and lower limits of agreement (with 95% CI), and the blue line represents the geometric mean – bias. DILI, drug-induced liver injury; ELISA, enzyme-linked immunosorbent assay; K18, cytokeratin 18; LFIA, lateral flow immunoassay; SERS, surface enhanced Raman scattering.

The following text has been added to the Discussion:

“For both studies there is good agreement between the K18 concentration obtained from the SERS-LFIA and ELISA tests for patients with DILI, with both producing high K18 concentrations. However, in non-DILI patients, the SERS-LFIA test reported higher K18 concentrations compared to the ELISA. We hypothesise that this occurred due to differences in assay protocols.³⁴ The ELISA involves multiple wash steps designed to remove non-specific binding, resulting in consistently low K18 concentrations in non-DILI samples. In contrast, the SERS-LFIA lacks these wash steps to enable rapid testing, which may allow non-specific proteins to remain and contribute to the overall signal. An additional factor that could contribute to the elevated K18 signal in non-DILI samples is the high density of antibodies on the nanoparticle surface used in the SERS-LFIA format. As noted by Sotnikov et al., steric

restrictions associated with densely immobilised antibodies can enhance antigen binding in the context of protein targets such as K18.³⁵ This may account for the greater antigen-antibody interactions observed in the SERS-LFIA compared to ELISA, particularly at lower K18 concentrations. Finally, the calibration curve was created by spiking K18 protein into healthy serum which will result in overestimate of K18 concentration at the lower range due to endogenous K18 in the healthy serum.”

Reviewer #4 (Remarks to the Author):

We are grateful to the reviewer for their addition input and careful reading of our manuscript.